# Non-Professional Phagocytosis Increases in Melanoma Cells and Tissues with Increasing E-Cadherin Expression

Luzie Helene Unseld [1,2,†], Laura S. Hildebrand [1,2,†], Florian Putz [1,2], Maike Büttner-Herold [3], Christoph Daniel [3], Rainer Fietkau [1,2] and Luitpold Valentin Distel [1,2,*]

1 Department of Radiation Oncology, Universitätsklinikum Erlangen, Friedrich-Alexander-Universität Erlangen-Nürnberg, Universitätsstr. 27, 91054 Erlangen, Germany; lu.unseld@gmail.com (L.H.U.); laura.hildebrand@uk-erlangen.de (L.S.H.); florian.putz@uk-erlangen.de (F.P.); rainer.fietkau@uk-erlangen.de (R.F.)

2 Comprehensive Cancer Center Erlangen-EMN, 91054 Erlangen, Germany

3 Department of Nephropathology, Institute of Pathology, Universitätsklinikum Erlangen, Friedrich-Alexander-Universität Erlangen-Nürnberg, 91054 Erlangen, Germany; maike.buettner-herold@uk-erlangen.de (M.B.-H.); christoph.daniel@uk-erlangen.de (C.D.)

* Correspondence: luitpold.distel@uk-erlangen.de

† These authors contributed equally to this work.

**Abstract:** Non-professional phagocytosis in cancer has been increasingly studied in recent decades. In malignant melanoma metastasis, cell-in-cell structures have been described as a sign of cell cannibalism. To date, only low rates of cell-in-cell structures have been described in patients with malignant melanoma. To investigate these findings further, we examined twelve primary melanoma cell lines in both adherent and suspended co-incubation for evidence of engulfment. In addition, 88 malignant melanoma biopsies and 16 healthy tissue samples were evaluated. E-cadherin levels were determined in the cell lines and tissues. All primary melanoma cell lines were capable of phagocytosis, and phagocytosis increased when cells were in suspension during co-incubation. Cell-in-cell structures were also detected in most of the tissue samples. Early T stages and increasingly advanced N and M stages have correspondingly lower rates of cell-in-cell structures. Non-professional phagocytosis was also present in normal skin tissue. Non-professional phagocytosis appears to be a ubiquitous mechanism in malignant melanoma. The absence of phagocytosis in metastases may be one reason for the high rate of metastasis in malignant melanoma.

**Keywords:** malignant melanoma; cell-in-cell structures; non-professional phagocytosis; cell cannibalism; E-cadherin; metastasis

## 1. Introduction

Cell-in-cell (CIC) structures in cancer have long been recognized in science. They were first described in 1891 by Dr. Julius Steinhaus. He mainly focused on breast cancer cells, because in these cell lines he observed the highest rate of CIC structures of all cancers he studied [1]. They are not new, but it is only in recent years that the focus has shifted to these structures. Many studies have been conducted on the prognostic factor of the appearance of cell-in-cell structures in carcinomas with mixed outcomes. In breast squamous cell carcinoma, it may be a useful diagnostic indicator [2–4]. On the other hand, cell-in-cell structures may be a poor prognostic factor in lung or rectal cancer [5–7].

Cell-in-cell structures are divided into several subgroups. Homotypic CIC is the engulfment of cells of the same type, contrary to heterotypic CIC wherein different cell types interact. Emperipolesis is the process by which a live leucocyte attaches to another cell and which engulfs the lymphocyte to form a cell-in-cell structure [3,8–10]. Entosis, on the other hand, describes the internalization of one still living cell by another living cell, which can be observed in breast cancer cells of the same type and the internalized cell then

undergoing cell death [3,8,9,11]. Another form of non-professional phagocytosis is cell cannibalism, in which living cells, especially cancer cells, feed on other cells, regardless of whether they are cancer or non-cancerous. This mechanism provides them with nutrients and ensures their survival [3,12,13]. When the cell signals to be eaten by another cell, it is called phagoptosis and it is a controlled form of cell death [14].

Malignant melanoma has a poor prognosis, especially when metastases are found, with sentinel lymph node status being the most important prognostic factor [15,16]. Therapeutic options for malignant melanoma consist of chemotherapy, immunotherapy and even radiation, especially for metastasis. However, with the number of deaths from melanoma still increasing, the research for new methods and therapies is still ongoing [15–18].

This study focuses on homotypic non-professional phagocytosis in vitro using twelve human malignant melanoma cell lines. Here, we try to identify whether cell-in-cell structures can be found. Furthermore, we compare the rate of CIC structures for the cell lines co-incubated and adherent in a cell flask with those co-incubated in suspension. In addition to the in vitro aspect, we also wanted to explore the in vivo behavior of human melanomas. Therefore, we analyzed 176 biopsies from 88 melanoma cases and 32 biopsies of normal tissue from 16 cases. In addition, our goal was to determine whether there was a relationship between the adhesion molecule E-cadherin and the rate of CIC.

## 2. Materials and Methods

### 2.1. Cell Lines and Cell Culture

Twelve different human melanoma cell lines were used in these experiments. The primary tumor cell lines ANST, ARPA, BIMA, ETFL, HV18MK, ICNI, ILSA, LIWE, and RERO were obtained from patients at the Department of Dermatology, University Hospital Erlangen, Germany. The patients gave consent and the Ethics Committee of the Medical Faculty of the Friedrich-Alexander-Universität Erlangen-Nürnberg approved the use (Ethic approval No.204_17 Bc). The established cell lines A375M, Mel624, and PMelL were purchased commercially. All cell lines were used for co-incubation in adhesion and in suspension.

### 2.2. CIC Induction in Adherent Cells

Depending on the growth rate of the melanoma cells, 60,000 to 500,000 cells were seeded on a coverslip and three times the number of seeded cells from the same cell line were placed in a cell culture flask. After cell culture flask and coverslip were incubated for 48 h at 37 °C in a 5% $CO_2$ atmosphere, the medium was removed, and the cells on the coverslip were washed with Dulbecco's Phosphate Buffered Saline (1xPBS, Sigma-Aldrich, Munich, Germany) and then stained with a solution of 1xPBS and CellTracker green CMFDA (Invitrogen, Auckland, New Zealand) for 20 min in the incubator. The staining solution was then removed and fresh melanoma medium was added and incubated for an additional 10 min. Finally, cells were washed three times for five minutes with 1xPBS. The cell culture flask was washed with 1xPBS after 48 h and then stained with CyTRAK orange (Thermo Fisher, Schwerte, Germany) for 30 min. After that, the medium was placed in a centrifuge tube and the remaining cells in the cell culture flask were detached with trypsin. Thereafter, all cells were detached from the bottom of the flask and transferred into the centrifuge tube. This tube was now treated with hyperthermia for 60 min in a 60 °C water bath. These hyperthermic cells were then added to the live cells on the coverslip and incubated again at 37 °C for 5 h.

After 5 h, the medium was removed from the coverslip and then washed with 1xPBS and then fixed with 37% formaldehyde (Merck, Darmstadt, Germany) in 1xPBS for 15 min. The coverslips were washed three times with 1xPBS for 5 min and blocked for at least 4 h in the refrigerator with a blocking solution consisting of 10 mL of 10% FBS, 1 g of bovine albumin, 90 mL of 1xPBS, and 0.3 mL of 1 M sodium acid. The coverslip was then washed three times with 1xPBS for 5 min and stained with 100 μL DAPI solution consisting of 3 μL DAPI (Roche, Grenzach-Whylen, Germany) and 10 mL 4xSSC/Tween for 1 min,

washed twice with distilled water, dried, and mounted on a slide with Vectashield + DAPI mounting medium (Vector Laboratories Inc., Burlingame, CA, USA).

### 2.3. CIC Induction in Suspended Cells

The process of CIC induction in suspended cells was essentially the same, except that the live cells were seeded in a culture flask rather than on a coverslip, then detached with trypsin and placed in a centrifuge tube. This centrifuge tube was centrifuged at 180 g for 8 min, the cells were resuspended with the medium, and the hyperthermic-treated cells were added. After 5 h of incubation, the cells were centrifuged onto a slide, fixed, blocked, and stained with DAPI. Finally, the coverslip was placed on the slide containing the DAPI-containing embedding medium.

### 2.4. Imaging and Image Analysis

The images were acquired using a semi-automated microscope (AxioImager Z2, Zeiss, Göttingen, Germany) and a software for image acquiring (Metasystems, Altlussheim, Germany). Prior to this, the areas were manually marked and the images were acquired automatically. Biomas image processing software (MSAB, Erlangen, Germany) was used for image analysis. The software automatically detected overlapping dead and viable cells and counted them as cell-in-cell structures. This process was then manually verified. In the 12 cell lines, 18,838/5509 (adherent/suspension) green-stained living cells, 2084/6794 red-stained dead cells, and 959/716 cell-in-cell structures were counted. The adherent cell sample has low cell counts because nonadherent or non-engulfed cells are lost during the wash steps (Figure 1C). GraphPad Prism 8 (GraphPad, San Diego, CA, USA) was used for statistical analysis.

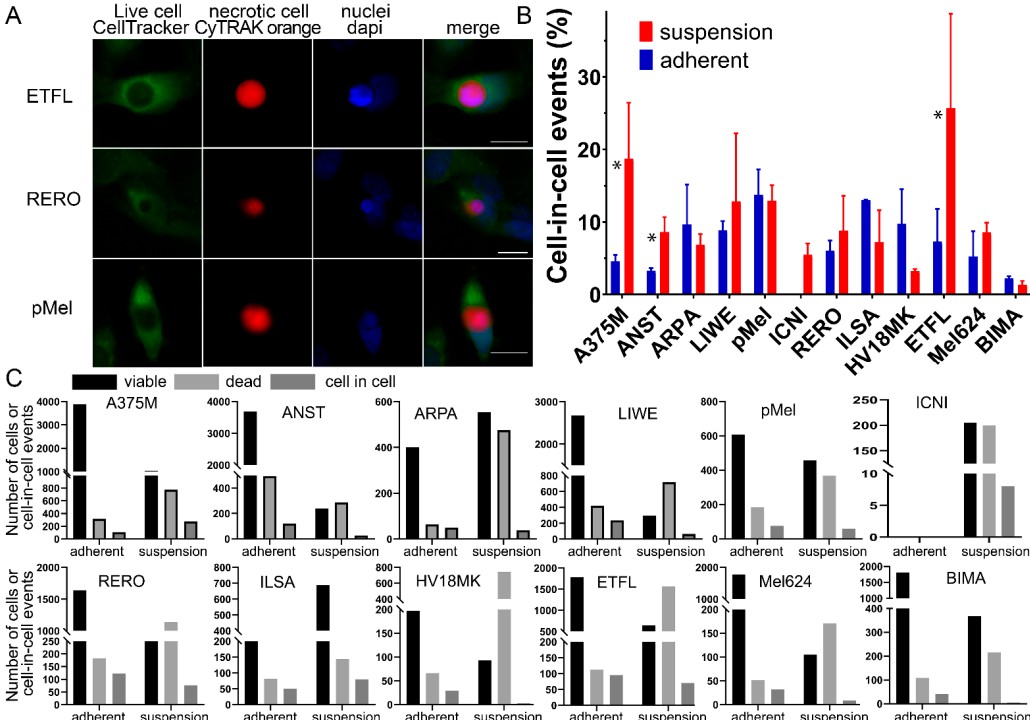

**Figure 1.** Cell-in-cell structures in melanoma cell lines in adhesion and suspension. (**A**) CIC structures in ETFL, RERO, and PMel stained with CellTracker Green CMFDA, CyTRAK orange and DAPI, fluorescent microscopy images. (**B**) All melanoma cell lines studied with either co-incubation in suspension or adhesion. Asterisks indicate significant difference between adherent and suspension cocultures. (**C**) Each melanoma cell line is compared to the amount of counted CIC structures (dark gray) versus the amount of viable cells (black) and the amount of dead cells (light gray). Asterisk indicate *p*-value ≤ 0.05.

*2.5. Criteria for CIC Evaluation*

To be counted as a cell-in-cell structure, a dead, red-stained cell had to be swallowed by a viable, green-stained cell. The nucleus of the viable cell was deformed by compression, leaving a semicircular imprint on one side of the nucleus. The software calculated a CIC rate as a percentage of the quotient of the CIC structures and all viable cells.

*2.6. CIC in Malignant Melanoma Tissues*

The commercial tissue microarray ME2081 containing 176 biopsies from 88 melanoma cases and 32 biopsies of skin normal tissue from 16 cases was purchased from BioCat (Heidelberg, Germany). The tissue spots were 1 mm in diameter. E-cadherin staining was performed on a BenchMark Ultra stainer (Roche, Grenzach-Wyhlen, Germany) using CC1 buffer (Benchmark ULTRA CC1, Roche) for antigen retrieval and an E-cadherin antibody (BD, Heidelberg, Germany). The average staining intensity was scored so that each sample received an independent score. A semiquantitative score from zero to three was assigned for the intensity of membranous E-cadherin. Intense E-cadherin staining was scored as high (3), moderate-to-high (2), low-to-moderate (1), and no staining intensity was scored as low (0). CIC structures were manually detected using image processing software.

**3. Results**

*3.1. Cell-in-Cell Structures in Cells under Adherent Versus Suspended Cells*

Living mammalian cells engulf dead cells by unprofessional phagocytosis, resulting in cell-in-cell (CIC) structures (Figure 1A). Typical of a CIC is that the dead cell is completely engulfed by the host cell and the nucleus of the host cell is crescent-shaped. The dead cell is circular because there is no way for it to adhere within the host cell. Twelve malignant melanoma cell cultures or cell lines were studied for their ability to phagocytose homotypic necrotic cells. All adherent cells and cells in suspension were able to form cell-in-cell structures, with the exception of the ICNI cell line, which could not be adherently cultured due to a lack of attachment to the cell bottle. Between the different adherent cell lines, we had a range of CIC structures from 2.2% in the BIMA cell line up to 13.0% in the adhesive culture of the ILSA cells. However, on average, more CIC structures were detected when the cells were in suspension (11.5 $\pm$ 6.0%) than when they were adherent (7.5 $\pm$ 3.9%) ($p$ = 0.034). In particular, the A375M and the ETFL cell lines formed much higher CIC rates in suspension compared to the adherent cells. A375M had a rate of 3.8% in adhesion and 26.4% in suspension or 5.3% versus 35.3% for ETFL. On the other hand, in the suspension co-incubation we have a much wider range starting with 1.4% again in the BIMA melanoma cell culture and ending with 25.7% in the ETFL cell line. However, in six of the cell lines, CIC rates for adhesion were very similar to those for suspension (Figure 1B,C). For the calculation of cell-in-cell structures, viable cells were compared with the number of engulfed cells. An average of 1712 viable cells and 189 CIC structures were counted in adherent cell experiments, and 459 viable and 60 CIC structures were counted in suspension experiments. The number of dead cells with an average of 190 in the adherent experiments and 566 in the experiments in suspension were also counted, but not considered for the calculation. The count in the individual cell lines varied greatly (Figure 1C).

*3.2. CIC Structures Depending on the E-Cadherin Expression*

Since adhesion between the cells is a prerequisite for phagocytosis, we wanted to determine whether there was a relationship between the adhesion molecule E-cadherin and the rate of CIC structures. Cells were immunostained with anti-E-Cadherin and $\alpha$ Tubulin. E-cadherin intensity was analyzed using fluorescence imaging and image analysis software (Figure 2A). In addition, a human normal tissue keratinocyte cell line, HaCat, was used. The HaCat cell line had the highest E-cadherin intensity of 12.9 gray levels. The ARPA melanoma cell line had only half the intensity of 5.7 gray levels. All the others had very low E-cadherin intensities ranging from ILSA with 2.8 to BIMA with only 0.5 gray levels (Figure 2B). To examine the relationship between E-cadherin intensity and CIC rates,

we correlated the two. For CIC rates with the adhesively grown cell lines, there was a tendency (r 0.45; *p* = 0.217) for higher E-cadherin expression to result in more CIC structures (Figure 2C). However, in the suspension experiments, the correlation was much weaker (r −0.29; *p* = 0.362) (Figure 2D). Then, we were interested in whether CIC structures could also be observed in malignant melanomas tissue.

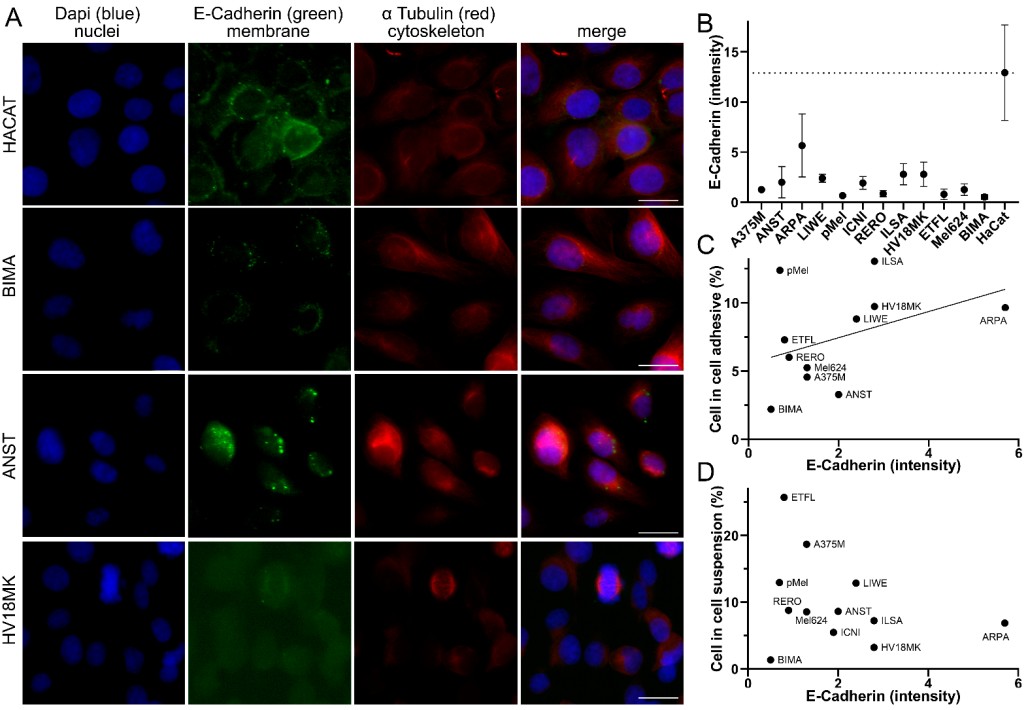

**Figure 2.** Correlation of cell-in-cell rates and E-cadherin intensity. (**A**) Malignant melanoma cell lines HACAT, BIMA, ANST, and HV18MK stained with DAPI, E-cadherin, and α-tubulin, fluorescence microscopy image. Scale bars are 20 μm. (**B**) E-cadherin intensity in grayscale of the analyzed melanoma cell lines. (**C**) Comparison of the intensity of E-cadherin with the percentage of CIC structures in the adhesive co-incubation of the melanoma cell lines analyzed. (**D**) E-cadherin intensity compared to percentage of CIC structures coincubated in suspension.

### 3.3. CIC Structures in Melanoma Biopsies Compared to TNM Stage

A total of 208 tissue specimens from 88 patients with malignant melanoma and skin samples from 16 healthy individuals were stained with anti-E-cadherin. This highlights the cell membranes for an easier detection of CIC structures (Figure 3A,B). In some samples, no CIC structures were found at all (Figure 3C). In normal tissue, only 28% showed CIC structures, whereas in T4 tumor, 43.5% of probes showed CIC structures. (Figure 3D). With increasing T stage, the incidence of CIC structures per square millimeter increased from no CIC structures found in the evaluated T2 melanoma biopsies to an average of 1.9 CIC structures/mm$^2$. In T4, the maximum CIC value reached 17.8 CIC/mm$^2$. No T1 stage melanoma were included, so no conclusion could be drawn regarding this stage (Figure 3E).

We also correlated the N stages with the frequency of CIC structures. In N0 stage biopsies, an average of 1.64 CIC structures/mm$^2$ was observed and cell-in-cell structures were found in 41% of N0 biopsies. In higher N stages, the numbers of CIC structures decreased. In N1 stage, CIC structures/mm$^2$ rates were 0.76 and CIC structures were present in 20% of biopsies. Finally, no CIC structures were found in N2 melanomas (Figure 3F). The results for stage M- are similar to the data for stage N-. The M0 biopsies show a rate of 1.56 CIC structures/mm$^2$. CIC structures were found in 39% of M0 melanomas. In metastatic melanoma, no CIC structures were found in the cores (Figure 3G). Cell-in-cell structures were also found in healthy skin tissue samples with rates of 1.15 CIC structures/mm$^2$. CIC structure counts in normal skin were intermediate between counts in T3 and T4 tumor, N0

and N1 stage, and M0 and M1 stage. There were no significant differences between CIC rates in N stages and metastases. We were then interested in whether CIC rates were also dependent on E-cadherin expression in the tissue.

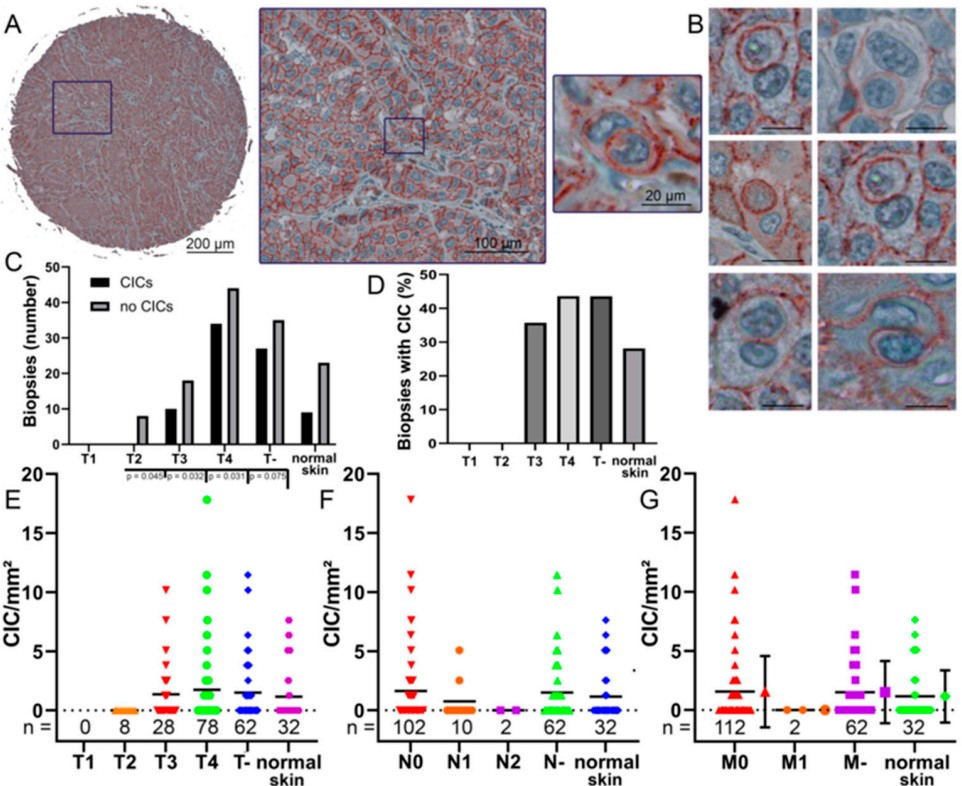

**Figure 3.** Cell-in-cell structures in melanoma biopsies. (**A**) A 1 mm spot of a tissue micro array and a cell-in-cell structure in a melanoma biopsy stained with anti-E-cadherin immunohistochemistry. (**B**) Multiple cell-in-cell structures in different melanoma biopsies stained with anti-E-cadherin. Scale bars are 20 µm. (**C**) Occurrence of CIC structures in the whole cores at different T stages compared to normal skin. T- indicating that the T stage is unknown. (**D**) Percentage of CIC structures in the whole cores at different T stages and normal skin. (**E**) The rate of CIC structures/mm$^2$ of each biopsy grouped by its T stage, T- indicating the T stage is unknown. (**F**) The rate of CIC structures/mm$^2$ of each biopsy grouped by its N stage, N- indicating the N stage is not known. (**G**) The rate of CIC structures/mm$^2$ of each biopsy compared to its M stage and its corresponding error bars, M-indicating the M stage is not known. *p* values are only given if this is smaller than *p* = 0.05. If no *p* values are given, then *p* values are present that are significantly larger. If there were less than 5 values in a group, no significance was calculated.

### 3.4. CIC Structures in Melanoma Tissues Are Dependent on the E-Cadherin Expression

E-cadherin intensity was categorized into four score levels ranging from no to high E-cadherin intensity (Figure 4A–D). Most tissues had low (33.7%) or low-to-moderate (35.1%) E-cadherin intensity. Moderate-to-high (23.6%) and high (7.7%) were much less common (Figure 4E). In tissues with no E-cadherin expression, CIC rates were lowest with only 0.29 ± 1.49 CIC structures/mm$^2$ and significantly lower compared to the other E-cadherin scores (*p* < 0.001). With increasing E-cadherin score, CIC rates increased to 3.8 ± 4.2 CIC structures/mm$^2$ for the high-intensity E-cadherin score (Figure 3F). At the lowest score, only 7.1% of the tissues showed CIC structures, whereas in score 3 E-cadherin intensity, more than 60% of the tissue cores had CIC structures (Figure 4G).

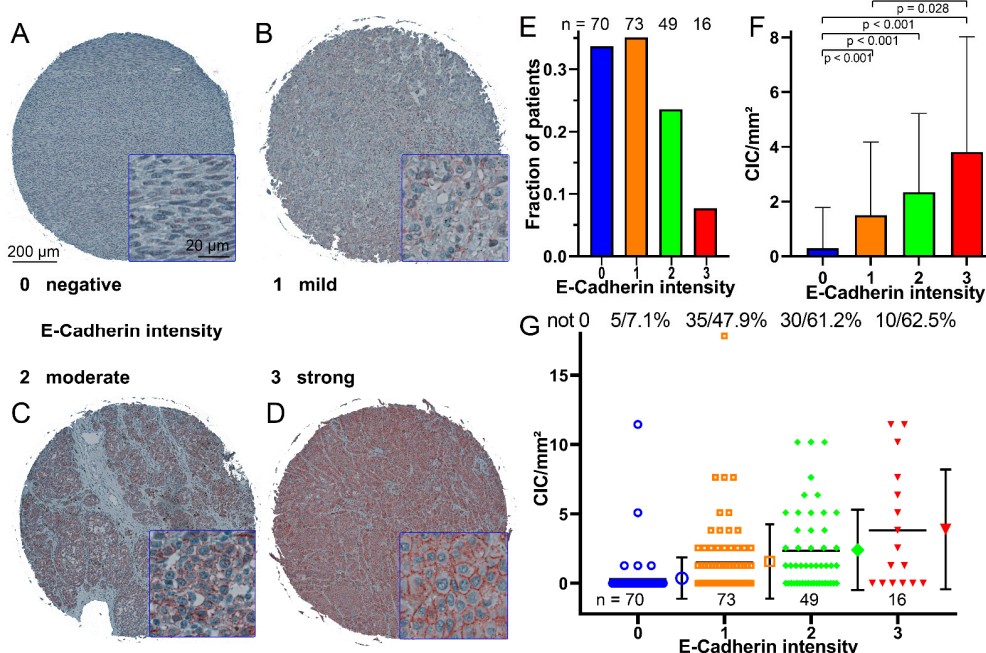

**Figure 4.** E-cadherin intensity in melanoma biopsies. (**A**) Example of a biopsy with little or no E-cadherin, immunohistochemically stained microscopic image stained with an E-cadherin antibody. (**B**) Example of a biopsy with low-to-moderate levels of E-cadherin, (**C**) Example of a biopsy with moderate-to-high levels of E-cadherin, (**D**) Example of a biopsy with high levels of E-cadherin. (**E**) Number of melanoma biopsies grouped by their E-cadherin intensity. (**F**) Occurrence of CIC structures/mm$^2$ in relation to the intensity of E-cadherin; the error bars indicate the standard deviation. (**G**) CIC structures/mm$^2$ rate of each biopsy grouped according to the intensity of E-cadherin, the mean value and the standard deviation is shown in the error bars. The number of cores that do not have 0 CIC structures and their percentage are given above the diagram.

## 4. Discussion

All of our 12 malignant melanoma cell lines were able to form cell-in-cell structures. Nine were low-passage number cell lines directly derived from patients, indicating that the patients' tumors were likely capable of non-professional phagocytosis. This suggests that non-professional phagocytosis is a common feature in malignant melanoma. In contrast to our results, non-professional phagocytosis was previously found only in cell lines derived from melanoma metastasis, but not in the primary tumor [19,20]. The experiments differ in the length of co-incubation time used, and we also co-incubated in suspension, which also lead to more cell-in-cell structures. Perhaps the longer co-incubation time increased the rate of CIC structures, as Lugini et al. also found an increase after 24 and 48 h of incubation. Furthermore, we only studied primary melanoma and not metastatic melanoma cells. Possibly even higher amounts of CIC events would have been observed in cell lines derived from metastatic melanoma. In addition, Lugini et al. used apoptotic cells induced by UVB radiation [19], whereas we used necrotic cell death induced by hyperthermia in our cells.

The higher CIC rates in suspension culture compared to adherent culture may be explained by the findings of Zhang et al. [21]. They reported that breast cancer cell cultures in suspension tended to have higher rates of organized actin when reattached to a surface, in contrast to adherent breast cancer cells, which had a higher incidence of unorganized, irregular actin. When the detached breast cells were planted on a surface, they reattached more rapidly and showed thicker and stronger stress fibers [21]. Actin plays an important role in cell cannibalism [8], so it can be assumed that the higher likelihood of stronger stress fibers in cells in the suspended state can lead to higher CIC rates. The cells can attach faster to the dead cells, allowing for more cell-in-cell structures to be detected. Overholtzer et al.

discovered similar findings with living cells and entosis. Due to the loss of attachment, the cells express a higher Rho-pathway activity, leading to higher engulfment rates [11].

What was impressive about the twelve cell lines was the large variance that occurred in CIC rates, indicating an inherent difference in the cell lines. Accordingly, we looked at the amount of E-cadherin expressed in the cells, which is known to decrease as tumors progress [22]. There was a correlation between high E-cadherin intensity and high cell-in-cell rates for cells in adhesion. In suspension experiments, however, this correlation did not exist. This suggests that E-cadherin is important for non-professional phagocytosis [23]. To test this hypothesis, experiments with knockdown and overexpression of E-cadherin in melanoma cell cultures could be performed to establish a causal relationship. However, other mechanisms must play a more important role, especially in suspension.

In contrast to the cell lines, CIC structures were often absent in malignant melanoma tissue cores. This may be due to several reasons. One would be that there really are no cell-in-cell structures. Second, cell-in-cell structures may have been overlooked. Moreover, in low cell-in-cell rates, no cell-in-cell structures may be sampled in the tissue. Lastly, CIC structures may only be seen temporarily, as the engulfed cells are degraded within approximately one day.

Another problem is that the biopsies were stained with anti E-cadherin to better visualize the cell wall, especially when a cell wall was found inside another cell wall, ergo a cell-in-cell structure. When little or no E-cadherin is present, cell-in-cell structures are more difficult to detect. However, even in the absence of E-cadherin, it is possible to detect cell-in-cell structures due to the stained nuclei and characteristic round cells. This could be a reason for the decrease in cell-in-cell structures [24,25]. However, we observed a decrease in the cell lines with decreasing E-cadherin. It would be likely that a decrease in E-cadherin in the tissue could also lead to a decrease of CIC structures in the tissue cores of malignant melanoma. If such studies are planned, consideration should be given to the availability of plasma membrane markers that are not expressed at reduced levels in advanced cancers. There are markers, such as CD138, that are relatively constant in expression [26]. Unfortunately, it is poorly expressed in malignant melanoma. Anti pan cadherin antibodies would be beneficial as they label N- and E-cadherin [27]. Other useful markers include Na/K ATPase and PMCA1 [28,29].

We did find a clear increase in CIC rates with increasing T stage. This is somewhat counterintuitive as one would expect a decrease in E-cadherin and a decrease in CIC rates in advanced T stages. A possible explanation may be that the thicker the melanoma, the higher the mitotic rate and growth rate [30]. T stage is defined by thickness, so a higher T stage is associated with a higher mitotic rate and a higher growth rate [31]. Higher N and M stages resulted in a clear decrease in CIC rates. This can be explained by a decrease in E-cadherin expression in metastatic malignant melanoma [24,25], which in turn causes the decrease in CIC rates [11].

CIC structures have also been found in normal tissues. This has already been described by Seeberger et al. who showed that many different normal tissue cell lines are capable of professional phagocytosis. He also showed that CIC events occur in untreated skin or in heart tissue in the necrotic area after myocardial infarction [32]. The occurrence of cell-in-cell structures has been demonstrated in a large number of cancers. They have been shown in head and neck cancer, breast cancer, pancreatic cancer, lung cancer, rectal cancer, anal cancer, and others [7,33–36]. It therefore appears that non-professional phagocytosis is an inherent property of almost all cell types [32,37]. In cell experiments, the only cells that could not be induced to ingest necrotic cells were B cell-derived lymphoblastoid cells [37]. This non-professional phagocytosis that we have studied is induced by necrotic cells. This allows for the rapid removal of necrotic cells without the need for professional phagocytic cells such as macrophages to migrate long distances to the necrotic cells [32]. In normal tissue, cells rarely die, so necrotic cells rarely occur and thus CIC structures are rarely observed. In cancers and their microenvironment with low nutrient supply, hypoxia, and acidity, cells die much more frequently and there is also a higher prevalence of necrotic death, so that CIC events

can be observed much more frequently in cancers [12,38]. In cancers, there are additional mechanisms that lead to increased non-professional phagocytosis [38]. Using cancer cell lines, it can be well demonstrated that very different phagocytosis rates occur, suggesting that different cancer properties lead to different phagocytosis rates [37,39]. Additionally, mediators such as IL-6 or IL-8 may be released, leading to increased CIC frequencies [40,41]. Experimentally, acidic cell culture conditions resulted in higher CIC rates. CIC can also lead to immune evasion by phagocytosing inflammatory cells or protecting ingested cells in host cells from inflammatory cells [3,20,42]. Thus, nonprofessional phagocytosis appears to be a general mechanism of almost all cells, which, moreover, can be upregulated in cancer and possibly downregulated, leading to improved cancer cell survival.

## 5. Conclusions

We found cell-in-cell events in a large proportion of malignant melanoma tissue sections and conclude that, in general, non-professional phagocytosis can occur in malignant melanoma. In metastatic melanoma, professional phagocytosis is reduced, which can be interpreted as an unfavorable prognostic factor, since cells that become detached from the tissue cannot be phagocytosed. This may be a component of the high metastatic rate of malignant melanoma. E-cadherin plays an important role for co-incubation in adhesion, but not in suspension.

**Author Contributions:** Conceptualization: L.V.D. and F.P.; methodology: L.V.D., L.H.U., L.S.H., C.D., M.B.-H. and L.S.H.; validation: L.V.D. and L.S.H.; formal analysis: L.H.U. and L.V.D.; investigation: L.H.U. and L.S.H.; resources: R.F.; data curation: L.V.D. and L.S.H.; writing—original draft preparation: L.H.U.; writing—review and editing: L.V.D., F.P., L.S.H., C.D. and M.B.-H.; visualization: L.H.U. and L.V.D.; supervision: L.V.D., L.S.H. and F.P.; project administration: L.V.D., F.P. and R.F. All authors have read and agreed to the published version of the manuscript.

**Funding:** We acknowledge financial support by Deutsche Forschungsgemeinschaft and Friedrich-Alexander-Universität Erlangen-Nürnberg within the funding program "Open Access Publication Funding".

**Institutional Review Board Statement:** The Ethics Committee of the Medical Faculty of the Friedrich-Alexander University of Erlangen-Nuremberg approved the production and use of the cell lines (Ethic approval No.204_17 Bc).

**Informed Consent Statement:** Written informed consent was obtained front door from all subjects involved in the study.

**Data Availability Statement:** The data presented in this study are available upon request from the corresponding author.

**Acknowledgments:** The present work was performed in fulfillment of the requirements for obtaining the degree "Dr. med".

**Conflicts of Interest:** The authors declare no conflict of interest.

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
