# Peer review of "Non-Professional Phagocytosis Increases in Melanoma Cells and Tissues with Increasing E-Cadherin Expression"

_curroncol, doi:10.3390/curroncol30080547_

Round 1

Reviewer 1 Report

This straightforward manuscript has examined cell-in-cell structures in cultured melanoma cells and human melanoma tissues, relating the latter to E-cadherin expression in the tumour cells. The study appears to have been performed very carefully and the presentation and discussion of results is very good. Overall, the manuscript is very well written, English language is excellent, the introduction and discussion sections have covered the subject with sufficient detail, and the findings of the study have not been over-interpreted. I have listed some minor English language corrections and some short questions and comments. Pending these minor corrections, I recommend this manuscript for publication.

1. The use of E-cadherin immunohistochemistry to assist in the detection of CIC structures was practical in this study, however there are more ubiquitously expressed membrane markers that may be more reliably applied for this purpose. While it is not necessary to repeat experiments for the present study, this could be added to the Discussion section, especially as the authors mention difficulties resulting from the variable expression of E-cadherin in cells and tissues.

2. Could the authors please keep the naming of the cell dyes consistent in section 2.2 and the Figure 1 legend.

3. In section 3.3, it is difficult to following the description of the results in the text because the number of melanoma specimens from each T or N stage is not given. Could these be added into the relevant figures (as it has been in some panels of Figure 4), or listed in the Methods section?

4. The final short paragraph in the Discussion section (lines 291-293) describing the identification of non-professional phagocytosis in normal tissues appears to have been added as an afterthought. Do the authors consider that these types of phagocytosis are an inherent property of cells, an adaptive mechanism that can be up- or down-regulated according to environmental or internal (intracellular) signals combined with the innate properties of cells? Or do they believe that the phenomenon is a hallmark of cancer that can be harnessed as a prognostic marker or therapeutic target?

Minor typographical errors

5. Line 44: ‘which engulfs’

6. Line 49: ‘whether’ (spelling)

7. Line 83: ‘Invitrogen’ (no hyphen)

8. Line 124: ‘malignant’

9. Line 153: ‘adhesion’

10. Line 166: ‘addition’

11. Line 202: ‘dependent’

12. Line 214: ‘are dependent on E-cadherin expression’

13. Line 235: (suggested alternative) “Nine were low passage number cell lines directly derived from patients”

14. Line 279: ‘cell-in-cell structures’  (The word ‘structures’ is missing).

15. The conventional term for the tissue pieces in tissue microarrays is ‘tissue cores’, not ‘tissue probes’. This could be changed throughout the document.

This manuscript is suitable for publication pending the minor corrections outlined in the comments to authors. Alternatively, the authors may wish to add more detail to their Discussion section.

Author Response

Dear editor, dear reviewers,

Thank you for the constructive criticism. We edited our manuscript according to your recommendations point by point. In the following, your comments are printed in italics and the insertions in the manuscript are printed normally.

Reviewer 1

This straightforward manuscript has examined cell-in-cell structures in cultured melanoma cells and human melanoma tissues, relating the latter to E-cadherin expression in the tumour cells. The study appears to have been performed very carefully and the presentation and discussion of results is very good. Overall, the manuscript is very well written, English language is excellent, the introduction and discussion sections have covered the subject with sufficient detail, and the findings of the study have not been over-interpreted. I have listed some minor English language corrections and some short questions and comments. Pending these minor corrections, I recommend this manuscript for publication.

  1. The use of E-cadherin immunohistochemistry to assist in the detection of CIC structures was practical in this study, however there are more ubiquitously expressed membrane markers that may be more reliably applied for this purpose. While it is not necessary to repeat experiments for the present study, this could be added to the Discussion section, especially as the authors mention difficulties resulting from the variable expression of E-cadherin in cells and tissues.

Yes, this is an urgent comment, and it should lead others to think more carefully about which membrane markers they use before they begin such an investigation. We only had one TMA and therefore could not take other stains as we noticed that some spots are not stained that well. Nevertheless, we had problems to identify really good markers during the literature search. We wrote the following: If such studies are planned, it should be considered whether plasma membrane markers are available that are not expressed at a reduced level in advanced tumors. There are markers such as CD138 that are constantly expressed [1]. Unfortunately, these are poorly expressed in malignant melanoma. Anti pan cadherin antibodies would be beneficial as it labels N- and E-cadherin[2]. Other useful markers include Na/K ATPase and PMCA1[3,4].

  1. Could the authors please keep the naming of the cell dyes consistent in section 2.2 and the Figure 1 legend.

We apologize for the different names. We have now used the same names in the figure, caption and M&M. In these experiments we used CellTracker Celltrace Green CMFDA and CyTRAK Orange. Since we used different red and green stains for different experiments, this was changed, but shouldn't happen. Thanks for noticing this problem!

  1. In section 3.3, it is difficult to following the description of the results in the text because the number of melanoma specimens from each T or N stage is not given. Could these be added into the relevant figures (as it has been in some panels of Figure 4), or listed in the Methods section?

You are absolutely right! We have included the number of patients in the figures so that it is much easier to realize the meaning of these graphs.

  1. The final short paragraph in the Discussion section (lines 291-293) describing the identification of non-professional phagocytosis in normal tissues appears to have been added as an afterthought. Do the authors consider that these types of phagocytosis are an inherent property of cells, an adaptive mechanism that can be up- or down-regulated according to environmental or internal (intracellular) signals combined with the innate properties of cells? Or do they believe that the phenomenon is a hallmark of cancer that can be harnessed as a prognostic marker or therapeutic target?

Thank you very much for this interesting question. We added to the discussion section following:

CICs have also been found in normal tissues. This has already been described by Seeberger et al. who showed that many different normal tissue cell lines are capable of professional phagocytosis. He also showed that CIC occurs in untreated skin or in heart tissue in the necrotic area after myocardial infarction [5]. The occurrence of cell-in-cell structures has been demonstrated in a large number of cancers. They have been shown in head and neck cancer, breast cancer, pancreatic cancer, lung cancer, rectal cancer, anal cancer and others [6-10]. It therefore appears that non-professional phagocytosis is an inherent property of almost all cell types [5,11]. In cell experiments, the only cells that could not be induced to ingest necrotic cells were B cell-derived lymphoblastoid cells [11]. This nonprofessional phagocytosis we studied is induced by necrotic cells. We see this as a protective function, that in tissue necrotic cells can be eliminated quickly without professional phagocytes like macrophages having to migrate long distances to the necrotic cell [5]. In normal tissue, cells rarely die, so necrotic cells rarely occur and thus CIC structures are rarely observed. In cancers with its microenvironment with low nutrient supply, hypoxia, and acidity, cells die much more frequently and among them also necrotic death, so that CIC events can be observed much more frequently in cancers [12,13]. In cancers, there are additional mechanisms that lead to increased non-professional phagocytosis [12]. Using cancer cell lines, it can be well demonstrated that very different phagocytosis rates occur, suggesting that different cancer properties lead to different phagocytosis rates [11,14]. Thus, mediators such as IL-6 or IL-8 may be released leading to increased CIC frequencies [15,16]. Experimentally, acidic cell culture conditions led to higher CIC rates. CIC can also lead to immune evasion by phagocytosing inflammatory cells or protecting ingested cells in host cells from inflammatory cells [17-19]. Thus, nonprofessional phagocytosis appears to be a general mechanism of almost all cells, which, moreover, can be upregulated in cancer and possibly downregulated, leading to improved cancer cell survival.

Minor typographical errors

  1. Line 44: ‘which engulfs’
  2. Line 49: ‘whether’ (spelling)
  3. Line 83: ‘Invitrogen’ (no hyphen)
  4. Line 124: ‘malignant’
  5. Line 153: ‘adhesion’
  6. Line 166: ‘addition’
  7. Line 202: ‘dependent’
  8. Line 214: ‘are dependent on E-cadherin expression’
  9. Line 235: (suggested alternative) “Nine were low passage number cell lines directly derived from patients”
  10. Line 279: ‘cell-in-cell structures’  (The word ‘structures’ is missing).
  11. The conventional term for the tissue pieces in tissue microarrays is ‘tissue cores’, not ‘tissue probes’. This could be changed throughout the document.

Many thanks for all the tips and improvements, which we have of course gladly implemented!!!

References:

  1. Gharbaran, R. Advances in the molecular functions of syndecan-1 (SDC1/CD138) in the pathogenesis of malignancies. Crit Rev Oncol Hematol 2015, 94, 1-17, doi:10.1016/j.critrevonc.2014.12.003.
  2. Billion, K.; Ibrahim, H.; Mauch, C.; Niessen, C.M. Increased soluble E-cadherin in melanoma patients. Skin Pharmacol Physiol 2006, 19, 65-70, doi:10.1159/000091972.
  3. Stauffer, T.P.; Guerini, D.; Carafoli, E. Tissue distribution of the four gene products of the plasma membrane Ca2+ pump. A study using specific antibodies. J Biol Chem 1995, 270, 12184-12190, doi:10.1074/jbc.270.20.12184.
  4. Rajasekaran, S.A.; Huynh, T.P.; Wolle, D.G.; Espineda, C.E.; Inge, L.J.; Skay, A.; Lassman, C.; Nicholas, S.B.; Harper, J.F.; Reeves, A.E., et al. Na,K-ATPase subunits as markers for epithelial-mesenchymal transition in cancer and fibrosis. Mol Cancer Ther 2010, 9, 1515-1524, doi:10.1158/1535-7163.MCT-09-0832.
  5. Seeberg, J.C.; Loibl, M.; Moser, F.; Schwegler, M.; Buttner-Herold, M.; Daniel, C.; Engel, F.B.; Hartmann, A.; Schlotzer-Schrehardt, U.; Goppelt-Struebe, M., et al. Non-professional phagocytosis: a general feature of normal tissue cells. Scientific reports 2019, 9, 11875, doi:10.1038/s41598-019-48370-3.
  6. Schenker, H.; Buttner-Herold, M.; Fietkau, R.; Distel, L.V. Cell-in-cell structures are more potent predictors of outcome than senescence or apoptosis in head and neck squamous cell carcinomas. Radiat Oncol 2017, 12, 21, doi:10.1186/s13014-016-0746-z.
  7. Schwegler, M.; Wirsing, A.M.; Schenker, H.M.; Ott, L.; Ries, J.M.; Buttner-Herold, M.; Fietkau, R.; Putz, F.; Distel, L.V. Prognostic Value of Homotypic Cell Internalization by Nonprofessional Phagocytic Cancer Cells. BioMed research international 2015, 2015, 359392, doi:10.1155/2015/359392.
  8. Breier, F.; Feldmann, R.; Fellenz, C.; Neuhold, N.; Gschnait, F. Primary invasive signet-ring cell melanoma. J Cutan Pathol 1999, 26, 533-536, doi:10.1111/j.1600-0560.1999.tb01802.x.
  9. Cano, C.E.; Sandi, M.J.; Hamidi, T.; Calvo, E.L.; Turrini, O.; Bartholin, L.; Loncle, C.; Secq, V.; Garcia, S.; Lomberk, G., et al. Homotypic cell cannibalism, a cell-death process regulated by the nuclear protein 1, opposes to metastasis in pancreatic cancer. EMBO molecular medicine 2012, 4, 964-979, doi:10.1002/emmm.201201255.
  10. Gottwald, D.; Putz, F.; Hohmann, N.; Buttner-Herold, M.; Hecht, M.; Fietkau, R.; Distel, L. Role of tumor cell senescence in non-professional phagocytosis and cell-in-cell structure formation. BMC Mol Cell Biol 2020, 21, 79, doi:10.1186/s12860-020-00326-6.
  11. Schwegler, M.; Wirsing, A.M.; Dollinger, A.J.; Abendroth, B.; Putz, F.; Fietkau, R.; Distel, L.V. Clearance of primary necrotic cells by non-professional phagocytes. Biology of the cell / under the auspices of the European Cell Biology Organization 2015, 107, 372-387, doi:10.1111/boc.201400090.
  12. Siquara da Rocha, L.O.; Souza, B.S.F.; Lambert, D.W.; Gurgel Rocha, C.A. Cell-in-Cell Events in Oral Squamous Cell Carcinoma. Front Oncol 2022, 12, 931092, doi:10.3389/fonc.2022.931092.
  13. Fais, S. Cannibalism: a way to feed on metastatic tumors. Cancer letters 2007, 258, 155-164, doi:10.1016/j.canlet.2007.09.014.
  14. Hofmann, A.; Putz, F.; Buttner-Herold, M.; Hecht, M.; Fietkau, R.; Distel, L.V. Increase in non-professional phagocytosis during the progression of cell cycle. PLoS One 2021, 16, e0246402, doi:10.1371/journal.pone.0246402.
  15. Ruan, B.; Wang, C.; Chen, A.; Liang, J.; Niu, Z.; Zheng, Y.; Fan, J.; Gao, L.; Huang, H.; Wang, X., et al. Expression profiling identified IL-8 as a regulator of homotypic cell-in-cell formation. BMB Rep 2018, 51, 412-417, doi:10.5483/BMBRep.2018.51.8.089.
  16. Wang, S.; Li, L.; Zhou, Y.; He, Y.; Wei, Y.; Tao, A. Heterotypic cell-in-cell structures in colon cancer can be regulated by IL-6 and lead to tumor immune escape. Exp Cell Res 2019, 382, 111447, doi:10.1016/j.yexcr.2019.05.028.
  17. Lugini, L.; Matarrese, P.; Tinari, A.; Lozupone, F.; Federici, C.; Iessi, E.; Gentile, M.; Luciani, F.; Parmiani, G.; Rivoltini, L., et al. Cannibalism of live lymphocytes by human metastatic but not primary melanoma cells. Cancer Res 2006, 66, 3629-3638, doi:10.1158/0008-5472.CAN-05-3204.
  18. Wang, X.; Li, Y.; Li, J.; Li, L.; Zhu, H.; Chen, H.; Kong, R.; Wang, G.; Wang, Y.; Hu, J., et al. Cell-in-Cell Phenomenon and Its Relationship With Tumor Microenvironment and Tumor Progression: A Review. Front Cell Dev Biol 2019, 7, 311, doi:10.3389/fcell.2019.00311.
  19. Gutwillig, A.; Santana-Magal, N.; Farhat-Younis, L.; Rasoulouniriana, D.; Madi, A.; Luxenburg, C.; Cohen, J.; Padmanabhan, K.; Shomron, N.; Shapira, G., et al. Transient cell-in-cell formation underlies tumor relapse and resistance to immunotherapy. Elife 2022, 11, doi:10.7554/eLife.80315.

Reviewer 2 Report

This study tried to determine whether there is a relationship between the adhesion molecule E-cadherin and the rate of Cell-in-cell (CIC). Could authors please make their figures in a right way before I can review it.

Figure 1a and 2a, author should also provide colorful pictures in other lane, not just overlap lane, such as green, blue and red.

Figure 1b and c, can author provide sd value and p value for these bar figures.

Figure 3c,d, any parallel experiments had been done? There is not sd value in these bar figures.

Figure 3e,3f,3g,4g, do these data have a statistic difference between them, if yes, please mark.

Moderate editing of English language required

Author Response

Dear editor, dear reviewers,

Thank you for the constructive criticism. We edited our manuscript according to your recommendations point by point. In the following, your comments are printed in italics and the insertions in the manuscript are printed normally.

Reviewer 2

This study tried to determine whether there is a relationship between the adhesion molecule E-cadherin and the rate of Cell-in-cell (CIC). Could authors please make their figures in a right way before I can review it.

Figure 1a and 2a, author should also provide colorful pictures in other lane, not just overlap lane, such as green, blue and red.

We have colored the previous black and white images with their corresponding colors.

Figure 1b and c, can author provide sd value and p value for these bar figures.

We have included error bars and p-values in Figure 1b. In the figure 1c no error bars are possible, because they are the listed and counted values, of which there is ultimately only one value.

Figure 3c,d, any parallel experiments had been done? There is not sd value in these bar figures.

Figure 3c is the number of biopsies with positive or negative cores. Since it is only one number, no error bar can be given here. Figure 3d shows the percentage of cores with CIC, again no error can be given. We tried to clarify this in the image caption.

Figure 3e,3f,3g,4g, do these data have a statistic difference between them, if yes, please mark.

We included p values, if they were smaller than 0.05. If there are no p values, than there are p values greater than 0.05. To clarify this, we included this in the caption.

In the caption: “p values are only given if this is smaller than p = 0.05. If no p values are given, then p values are present that are significantly larger. If there were less than 5 values in a group, no significance was calculated.”

Figure 4G has the same p values as Figure 4F and only depicts the single cores/individuals. It depicts, which fraction of the cores with the different E-cadherin intensities that are different from 0 CIC. This is given in the superscript of Figure 4G.

In the caption was included: The number of cores that do not have 0 CIC and their percentage are given above the diagram.

Reviewer 3 Report

Authors: Unseld L., Hildebrand L.S., Putz F., et al.  

Title: "Non-professional phagocytosis increases in melanoma cells ..." 

COMMENTS:  

The Authors explored the phenomenon of non-professional phagocytosis (cell-in-cell or CIC) in melanoma cells in vitro and in vivo and also interrelation of this phenomenon with the expression of E-cadherin. They demonstrate the revealed correlation between CIC and increase in the E-cadherin level in melanoma cell cultures and biopsy samples. The manuscript is well written and illustrated. 

Criticism: 

1. It seems strange that the Authors did not try to connect their findings to pathogenesis of melanoma. In particular, they could study how CIC correlates with signaling pathways and/or expression of genes (different of E-cadherin) responsible for aggressive growth of melanomas, metastases, resistance to therapeutics, etc. At least, it would be interesting to explore how CIC correlates with melanoma cell migration, melanoma cell resistance to apoptosis, etc. 

2. After revealing the correlation between CIC and E-cadherin expression, the Authors could try to perform experiments with knockdown and overexpression of E-cadherin in melanoma cell cultures to establish a causal link. Why did they not perform such experiments? 

Author Response

Dear editor, dear reviewers,

Thank you for the constructive criticism. We edited our manuscript according to your recommendations point by point. In the following, your comments are printed in italics and the insertions in the manuscript are printed normally.

Reviewer 3

The Authors explored the phenomenon of non-professional phagocytosis (cell-in-cell or CIC) in melanoma cells in vitro and in vivo and also interrelation of this phenomenon with the expression of E-cadherin. They demonstrate the revealed correlation between CIC and increase in the E-cadherin level in melanoma cell cultures and biopsy samples. The manuscript is well written and illustrated. 

Criticism: 

  1. It seems strange that the Authors did not try to connect their findings to pathogenesis of melanoma. In particular, they could study how CIC correlates with signaling pathways and/or expression of genes (different of E-cadherin) responsible for aggressive growth of melanomas, metastases, resistance to therapeutics, etc. At least, it would be interesting to explore how CIC correlates with melanoma cell migration, melanoma cell resistance to apoptosis, etc.

Our primary goal of our work was actually only to determine whether and how frequently CIC structures occur in malignant melanoma cell lines and cell tissues. Then we noticed the association with E-cadherin and studied this further. We fully agree that it would be very interesting to study the correlation between the aggressive growth of malignant melanoma and the occurrence of CIC. However, this would be a completely different work that our group could hardly do. We think that we could show some interesting aspects and hope that other research groups can build on these aspects.

  1. After revealing the correlation between CIC and E-cadherin expression, the Authors could try to perform experiments with knockdown and overexpression of E-cadherin in melanoma cell cultures to establish a causal link. Why did they not perform such experiments? 

Yes, we too believe that these would be extremely interesting aspects and would provide much better evidence of the importance of E-cadherin expression. However, as mentioned earlier, this is outside the scope of what we can perform. Therefore, we can only hope that such experiment will be performed by others.

We include your idea in the discussion section: To test this hypothesis, experiments with knockdown and overexpression of E-cadherin in melanoma cell cultures could be performed to establish a causal relationship.

Round 2

Reviewer 3 Report

Dear Authors, 

I am satisfied by your response and explanations. 

Author Response

Thank you very much for the detailed review!